# Genome-Wide Identification, Evolution, and Expression Analysis of the WD40 Subfamily in *Oryza* Genus

**DOI:** 10.3390/ijms242115776

**Published:** 2023-10-30

**Authors:** Simin Ke, Yifei Jiang, Mingao Zhou, Yangsheng Li

**Affiliations:** State Key Laboratory of Hybrid Rice, College of Life Sciences, Wuhan University, Wuhan 430072, China; 2021202040080@whu.edu.cn (S.K.); 2021102040039@whu.edu.cn (Y.J.); 2021202040077@whu.edu.cn (M.Z.)

**Keywords:** rice, *Oryza* genus, WD 40, expression patterns, gene family

## Abstract

The WD40 superfamily is widely found in eukaryotes and has essential subunits that serve as scaffolds for protein complexes. WD40 proteins play important regulatory roles in plant development and physiological processes, such as transcription regulation and signal transduction; it is also involved in anthocyanin biosynthesis. In rice, only *OsTTG1* was found to be associated with anthocyanin biosynthesis, and evolutionary analysis of the WD40 gene family in multiple species is less studied. Here, a genome-wide analysis of the subfamily belonging to WD40-TTG1 was performed in nine AA genome species: *Oryza sativa* ssp. *japonica*, *Oryza sativa* ssp. *indica*, *Oryza rufipogon*, *Oryza glaberrima*, *Oryza meridionalis*, *Oryza barthii*, *Oryza glumaepatula*, *Oryza nivara*, and *Oryza longistaminata*. In this study, 383 WD40 genes in the *Oryza* genus were identified, and they were classified into four groups by phylogenetic analysis, with most members in group C and group D. They were found to be unevenly distributed across 12 chromosomes. A total of 39 collinear gene pairs were identified in the *Oryza* genus, and all were segmental duplications. WD40s had similar expansion patterns in the *Oryza* genus. Ka/Ks analyses indicated that they had undergone mainly purifying selection during evolution. Furthermore, WD40s in the *Oryza* genus have similar evolutionary patterns, so *Oryza sativa* ssp. *indica* was used as a model species for further analysis. The cis-acting elements analysis showed that many genes were related to jasmonic acid and light response. Among them, *OsiWD40-26/37/42* contained elements of flavonoid synthesis, and *OsiWD40-15* had MYB binding sites, indicating that they might be related to anthocyanin synthesis. The expression profile analysis at different stages revealed that most *OsiWD40s* were expressed in leaves, roots, and panicles. The expression of *OsiWD40s* was further analyzed by qRT-PCR in 9311 (indica) under various hormone treatments and abiotic stresses. *OsiWD40-24* was found to be responsive to both phytohormones and abiotic stresses, suggesting that it might play an important role in plant stress resistance. And many *OsiWD40s* might be more involved in cold stress tolerance. These findings contribute to a better understanding of the evolution of the WD40 subfamily. The analyzed candidate genes can be used for the exploration of practical applications in rice, such as cultivar culture for colored rice, stress tolerance varieties, and morphological marker development.

## 1. Introduction

WD40, also known as the WD-repeat (WDR) protein, is a family of proteins that have been highly conserved during evolution, and they are widely found in eukaryotes [1]. The WD40 domain is generally 4–8 amino acid residues of about 40–60, containing a glycine–histidine dimer peptide (GH) at the N-terminal end and a tryptophanaspartate dimeric peptide (WD) at the C-terminal end [2,3]. Typically, the WD40 domain has a β-propeller structure containing 4–9 blades, and its scaffold-type structure can serve as an interaction site for a variety of biomolecules, including proteins, peptides, RNA, and DNA, which helps proteins exert their biological activities [4,5,6]. WD40 proteins have complex structures and functions; they can interact with diverse proteins in various ways and are involved in many bioregulatory processes such as DNA replication, damage response, transcriptional regulation, post-translational modification, signal transduction, and apoptosis [7,8,9,10]. In addition, most WD40 proteins are found in eukaryotes, and only a small fraction is found in prokaryotes [11]. In lower organisms, WD40 is mainly involved in cell cycle, growth, and development, while in higher organisms, it plays an important role in cell cycle, apoptosis, signal transduction, abiotic stress, and other physiological and biochemical processes [12,13,14].

Previously, the researchers classified WD40 proteins in different species based on evolutionary relationships and sequence similarity. Due to low sequence conservation and high functional diversity of the WD40 family, 237 potential *AtWD40* proteins containing at least 4 WDR motifs were identified in *Arabidopsis thaliana* and classified into 33 groups [15]; in the peach genome, 220 WD40 superfamily members were identified, and the *MtWD40s* were further divided into five subfamilies based on grouping in *Arabidopsis thaliana* [16]; physiological biochemistry, genetic structure and conserved protein motifs of 42 *JrWD40s* were identified and analyzed in walnut [17]; in japonica rice, a total of 200 members of WD40 were identified and categorized into 11 subfamilies based on structural domain features [18]. A specialized database, WDSP “http://wu.scbb.pkusz.edu.cn/wdsp/ (accessed on 20 June 2023)”, has been developed to provide detailed information on WD40-repeat proteins of individual species. WDSPdb contains 63,211 WD40-repeat proteins identified in 3383 species, including most model organisms, and it can be used to view structural models, hotspot residues, etc. [19].

WD40 proteins generally have additional structural domains to recruit other factors to form protein–protein complexes [20]. Therefore, the WD40 protein family participates in many plant development and physiological processes and regulates the assembly of multiple complexes in plant resistance and abiotic stress response [21,22,23]. For example, in wheat (*Tritivum aestivum* L.), *TaWD40D* encoding seven WD40 domains was identified, which was a positive regulator of plant responses to salt stress and osmotic stress. The down-regulation of it resulted in decreased relative water content and impaired growth [24]. In tomato (*Solanum lycopersicum*), *SlWD40* was strongly co-expressed with master regulators such as RIN (ripening inhibitor) and NOR (nonripening). And *SlWD40* overexpression and RNAi lines exhibited substantially accelerated and delayed phenotypes compared with the wild type, respectively [25]. In rice, *OsLIS-L1* containing WD40 motifs played an important role in male gametophyte formation and the first internode elongation [26].

The most widely known function of WD40 is as part of the MYB-bHLH-WD40 (MBW) complex, involved in the regulation of anthocyanin/proanthocyanin biosynthesis. Anthocyanins are antioxidants used as natural colorants and are beneficial to human health, which contributes to reactive oxygen species detoxification and sustains plant growth and development [27]. The first WD40 protein related to anthocyanins biosynthesis identified in the model plant Arabidopsis was TRANSPARENT TESTA GLABRA1 (TTG1). *AtTTG1* encodes 341 amino acids and can form an MBW ternary complex with *AtMYB123* and *AtbHLH42* to synergistically activate the expression of BANYULS (anthocyanidin reductase) expression and promote the biosynthesis of proanthocyanin [28]. In contrast, the codon of the *Atttg1* mutant is terminated early, the *AtTTG1* protein is truncated at the far C-terminus, and the seedlings are unable to accumulate anthocyanins [29]. *Oryza sativa* ssp. *japonica OsTTG1*, *Vitis vinifera* (grape) *VvWDR1*, and *Camellia sinensis CsWD40* encode 355, 344, and 342 amino acids, respectively. In the *Osttg1* mutant of the DLMM rice variety, there was almost no anthocyanin accumulation in the rice bast, and the pericarp was nearly white, with only the abaxial side brown [30]. Overexpression of TTG1-like *CsWD40* in tobacco resulted in a significant increase in anthocyanin content in petals of transgenic plants; both anthocyanin and proanthocyanin content were increased when co-expressed with *CsMYB5e* in tobacco [31]. *VvWDR1* increases anthocyanin accumulation in transgenic tobacco through interaction with the *VvMYBA2r*-*VvMYCA1* complex [32].

In addition to anthocyanin biosynthesis, studies have shown that TTG1 is also related to seed coat mucilage biosynthesis [33], root–hair pattern [34,35], flowering time regulation [36], fruit spine initiation [37,38]. Among other processes, TTG1 might also directly or indirectly regulate downstream target genes through interactions with other proteins [39]. Current studies related to the regulation of TTG1 are more limited, and most of them dissociate the TTG1-containing MBW complex for regulation. For example, EIN3 and RSL4 interfere with WER-GL3-TTG1 (MBW complex) and mediate ectopic root hair formation in Arabidopsis [40], and SnRK1 inhibits anthocyanin biosynthesis through transcriptional regulation of the MYB-bHLH-TTG1 complex [41].

With the completion of the wild rice genome map, the *Oryza* genus is increasingly becoming an ideal system for evolutionary and functional genomics studies in gramineous plants. The AA genome is the genome type in the *Oryza* genus that contains the largest number and widest distribution of diploid species. The AA genome *Oryza* species is widely distributed in both natural and wild environments and contains many genes that can improve rice yield and stress tolerance [42,43]. However, there is no evolutionary analysis of the WD40 protein family in the *Oryza* genus. Therefore, the AA genome *Oryza* species was chosen to analyze the evolutionary relationships of the WD40 protein family. Since the WD40 protein family is enormous, this study will focus on the subfamily where *OsTTG1* is located and identify and analyze the WD40 subfamily in the rice AA genome, including *O. sativa japonica*, *O. sativa indica*, *O. longistaminata*, *O. barthii*, *O. glumaepatula*, *O. meridionalis*, *O. nivara*, *O. glaberrima* and *O. rufipogon*. Based on rigorous phylogenetic analysis, subfamily members were identified, and evolutionary relationships were inferred. In addition, 200 *OsWD40s* proteins were identified in japonica rice in previous studies [18], and the *Oryza* genus can be characterized similarly. Therefore, indica was chosen as a model plant for the analysis of cis-acting elements and expression patterns. The results provide a global view of the WD40-TTG1 subfamily in the AA genome *Oryza* genus, and candidate genes related to anthocyanins can be used to cultivate colored rice or develop morphological markers.

## 2. Results

### 2.1. Identification of WD40 Repeat Protein Family in Oryza Genus

In the reported study, a total of 200 members in japonica OsWD40s were identified, of which 50 members were in the subfamily where *OsTTG1* is located [18]. These 50 protein sequences were compared as queries in the remaining eight *Oryza* genera. After removing the duplicate sequences, a total of 383 members of the subfamily in which WD40-TTG1 is located were identified, including *O. japonica*, of which 51 in *O. sativa indica*, 9 in *O. longistaminata*, 48 in *O. barthii*, 46 in *O. glumaepatula*, 37 in *O. meridionalis*, 48 in *O. nivara*, 46 in *O. glaberrima*, 48 in *O. rudipogon*, 50 in *O. sativa japonica* (Figure 1a, Appendix A). To understand their evolutionary relationships, we constructed a phylogenetic tree and divided it into four groups (Figure 1b). Among them, groups C and D had the most members, with 113 and 123 members, respectively. *OsTTG1* (*LOC_Os02g45810*) was in group C, and indica *OsR498G0204537800.01* was close to it.

### 2.2. Characterization and Phylogenetic Relation of OsiWD40 Family Members in Indica

To understand the overall distribution of WD40s on the genome, the gene distribution of family members was visualized. Additionally, the current genomic information of *Oryza longistamata* is incomplete. Although the genome sequence has been sequenced, it has not been annotated and analyzed, so the chromosome distribution cannot be analyzed yet. Except for *Oryza longistaminata*, the distribution pattern of genes in the rest of the *Oryza* genus was similar (Figure 2a). They were widely and unevenly distributed on 12 chromosomes, and the highest numbers were found on their first five chromosomes, with fewer members in the remaining chromosomes. Next, chromosome distribution in indica was visualized, and its gene names were renamed in order (Figure 2b). The correlation coefficient between chromosome length and family gene numbers is 0.79, indicating that gene distribution showed a positive correlation with chromosome length.

In addition, some physicochemical properties of WD40s were predicted (Appendix A). WD40 gene members in *O. longistaminata* were fewer than other *Oryza* genera, but their amino acid numbers and molecular weight size range were wider than the rest of them, and the isoelectric point value changes were smaller. This might be due to its relatively homogeneous genetic background, and its proteins have diverse functions and structures that can adapt to different environmental pressures. Its isoelectric point range was small, indicating that most of its proteins carry negative charges, which might be related to its salt tolerance. Furthermore, the similarity of physicochemical properties of other *Oryza* genera suggested that they might have similar biochemical functions and structures. For example, in indica, the size of amino acids ranged from 72 (*OsiWD40-49*) to 888 (*OsiWD40-36*), and the predicted molecular weight ranged from 7861.78 (*OsiWD40-49*) to 97,991.2 (*OsiWD40-36*). The theoretical isoelectric points ranged from 4.3 (*OsiWD40-26*) to 9.81 (*OsiWD40-49*), with 32 acid isoelectric points and 19 basic isoelectric points. Subcellular localization predictions showed that 292 WD40s proteins were localized in the nucleus, 67 in the chloroplast, 15 in the extracellular, and 8 in membrane systems and mitochondria. This finding implies that the subfamily might be primarily involved in intranuclear developmental processes.

### 2.3. Analysis of Gene Structure and Conserved Motifs

The evolution of protein families is mainly manifested by the diversity of gene structures and changes in conserved motifs. To better understand the structure of WD40 proteins in the *Oryza* genus, ten conserved motifs were analyzed using MEME software (v5.5.4), and their domains were validated using the CDD database (Appendix A). The WD40 subfamily proteins in the *Oryza* genus were interspersed in the phylogenetic tree. Specifically, based on the similarity of protein sequences, most WD40 proteins in the *Oryza* genus had matching homologs and shared similar conserved motifs and structural domains. For ease of observation, *OsiWD40s*, members of the WD40 family in *O. sativa indica*, were extracted for structural characterization (Figure 3). The phylogenetic tree was constructed based on their protein sequences (Figure 3a), which can be divided into three subgroups. The conserved motif analysis showed that all *OsiWD40s* contained at least five or more motifs (Figure 3b). After performing the Tomtom algorithm search, motifs 1–4 and 9 all had Trp-Asp (WD) repeats signatures (Appendix A). Among them, most of the proteins in subgroups I and III had similar motif types and numbers, while subgroup II mostly contained 10 conserved motifs. This might indicate that the protein functions in subgroup II were more distinct from those in the other two groups. Structural domain analysis of these family members showed that each member has either a WD40 structural domain or a WD40 superfamily structural domain and other specific structural domains in different subgroups (Figure 3c). In addition, analysis of the gene structure of *OsiWD40s* showed that *OsiWD40s* had relatively more introns, with most members having more than five introns (Figure 3d). While genes of the same subgroup share similar exon/intron structures, group I had relatively longer introns, while group II and group III had relatively shorter introns. These findings contribute to the understanding of the structural diversity of the WD40 subfamily.

### 2.4. Collinearity Analysis of OsiWD40s in Oryza Genus

Gene duplication events are a critical part of gene family composition. Analysis of family gene duplication events with the Multiple Covariance Scanning (MCScanX) tool showed that, except *O. longistaminata*, in the remaining eight species *O. barthi*, *O. glaberrima*, *O. glumaepatula*, *O. sativa indica*, *O. meridionalis*, *O. nivara*, *O. rufipogon*, and *O. sativa japonica* a total of 6, 6, 5, 5, 3, 3, 5, and 6 collinear gene pairs were found (Table 1). The results showed that the duplication type of all collinear gene pairs was segmental/WGD duplication, and the gene pairs all showed a similar distribution in the genome, indicating that the expansion pattern of the WD40s was similar in multiple *Oryza* genera (Figure 4).

The Ka/Ks values of collinear gene pairs were used to assess the selection pressure of gene duplication. In general, Ka/Ks <1, =1, and >1 indicate that the genes are under purifying, neutral, and positive selection accordingly. The Ka/Ks values of all covariate gene pairs in the *Oryza* genus were less than 1, indicating that these genes had undergone strong purifying selection during evolution. In addition, the occurrence of replication events was calculated to range from 4.48 to 155.10 Mya for the gene pairs of eight *Oryza* genera.

### 2.5. Analysis of Cis-Regulatory Elements (CREs) in the Promoters of OsiWD40s

In recent years, with the advancement of high-throughput sequencing technology and the completion of sequencing of various plant genomes, the analysis of promoter sequences and cis-elements has played a role in predicting the function of regulatory genes. In order to understand the genetic function, metabolic network, and regulatory mechanism of WD40, the cis-acting elements of the promoter region can be analyzed. And, from the previous analysis, the physiological characteristics and structure of the *Oryza* genus were similar; indica was used as a model plant to analyze its cis-acting elements. The 2000 bp upstream sequence of *OsiWD40s* was taken as the hypothetical promoter, and its distribution and function were analyzed and visualized (Figure 5a). A total of 46 elements were identified after collating the structures, which can be divided into four major categories: abiotic, elements related to drought and low-temperature responses; hormone-related, elements related to plant hormones such as MeJA and abscisic acid; development-related, elements specifically expressed in various organ tissues; and light responsive elements (Figure 5c–e). *OsiWD40s* had a larger variety of elements associated with light responsive, while the variety of elements associated with hormone-related and abiotic was small but numerous (Figure 5b). Almost all members contained hormone-related cis-regulatory elements, suggesting that the expression of *OsiWD40s* genes might be responsive to various plant hormones. Furthermore, the cis-element of *OsiWD40-26*, *37*, and *42* were shown to be associated with flavonoid biosynthetic gene regulation. And *OsiWD40-15* had MYB binding sites and 11 hormone-related elements, including MeJA and gibberellin. The protein interaction analysis of OsiWD40-15 also revealed that it interacted with four bHLH proteins, as well as two proteins containing HTH myb-type structural domains (Appendix A), showing that it might be involved in anthocyanin regulatory pathways as a member of the MBW complex.

### 2.6. Expression Patterns of OsiWD40s Genes in Different Tissues

To verify the function of *OsiWD40s* identified in indica (R498), expression profile data were obtained from the MBK database, the expression data of OsiWD40s in tissues and organs at different periods were analyzed, and a heat map was drawn based on their expression (Figure 6). There were three spatial-temporal expression patterns of *OsiWD40s*: the first category, *OsiWD40-8/12/44/48/51*, was expressed in panicle, spike and flower, with lower expression at root and leaf; the second category (e.g., *OsiWD40-4/27*), in contrast to the first category, had high expression at root and leaf and low expression in pollen; the third category (e.g., *OsiWD40-14/18*) was expressed in all tissues and organs at specific times, with constitutive expression. In addition, collinear gene pairs such as *OsiWD40-5/OsiWD40-38* had similar expression patterns, both expressed in roots, leaves, etc., and low expression in pollen, suggesting that some *OsiWD40s* genes are functionally redundant. In contrast, some collinear gene pairs, such as *OsiWD40-12/OsiWD40-29*, had opposite expression patterns, with the former being expressed in pollen and the latter in roots and leaves, suggesting that some genes might have been functionally divergent during the evolutionary process. *OsiWD40-15* belonged to the third category of genes and was expressed in roots, leaves, flowers, and spikes, with lower expression in pollen. Moreover, the prediction of *OsiWD40s* protein–protein interactions revealed that *OsiWD40-25* was located at the center of the protein interaction network and is expressed in various tissues, suggesting that it might be involved in complex physiological processes. In brief, the specific expression pattern of OsiWD40s genes indicates that gene function might have evolved in response to various environmental conditions.

### 2.7. Expression Analysis of the OsiWD40s Genes in Response to Hormone and Abiotic Treatments

Then, the expression of OsiWD40s was further investigated under different phytohormonal/abiotic stresses, including gibberellin, jasmonic acid, auxin, and abscisic acid, as well as salinity and drought treatments. Fourteen genes in each subgroup of OsiWD40s were selected proportionally based on the number of members, and the expression pattern of OsiWD40s in leaves was examined by qRT-PCR. All 14 genes selected showed some differences in the degree of response to different treatments. After drought treatment, *OsiWD40-14*/*24/42* were more significantly up-regulated, while *OsiWD40-5/16/33/46* were significantly down-regulated; after salt treatment, *OsiWD40-4/14/24/30/33/42* were significantly up-regulated, while *OsiWD40-5* and *46* were significantly down-regulated. After cold treatment, *OsiWD40-4/5/14/19/30/42/46* were significantly up-regulated, and *OsiWD40-16/24/50* were down-regulated (Figure 7a). Above all, it can be speculated that *OsiWD40-14* and *OsiWD40-14* might play an important role in drought, salt, and cold stress. After hormone treatment, *OsiWD40-24* was significantly up-regulated by all treatments except for GA treatment, which was not significantly up-regulated (Figure 7b), and no gibberellin-related element was found in its cis-acting element prediction. In contrast, *OsiWD40-16* was significantly up-regulated after GA treatment, implying that *OsiWD40-24* and *OsiWD40-16* might be involved in hormone-related regulation. Taken together, *OsiWD40-24* responded to both hormone and abiotic stresses to a greater extent in the leaf, which was consistent with containing multiple abiotic stress and hormone response-related elements. What is more, many OsiWD40s were also more responsive to cold stress and might play a greater role in cold tolerance.

## 3. Discussion

The WD40 protein family is a large and diverse superfamily whose repeats, WD40, encode proteins containing a glycine-histidine (GH) pair at the N-terminal end and a tryptophan-aspartate (WD) pair at the C-terminal end, which play important roles in a wide range of physiological functions including growth, cell cycle, signal transduction, chromatin remodeling, and transcriptional regulation [44]. The WD40 repeat sequence is presumed to be caused by genomic duplication events and recombination [3,45].

In previous studies, 237, 200, 743, and 225 *WD40* genes were identified in *Arabidopsis*, *japonica*, wheat, and *foxtail millet*, respectively [18,46,47]. In addition, 161 WD40 subfamily DWD proteins were identified in soybeans [48]. The presence of abundant *WD40* genes in these plants indicates an evolutionary amplification of this superfamily. In addition to higher eukaryotes, such as rice, *WD40* has been found in prokaryotes and lower organisms [12,49]. To date, the cloning and functional studies of WD40 genes in rice have been limited. It has been found that rice pollen germination regulator (GORI) encodes a WD40 protein essential for rice pollen growth [50,51], the SRWD1-WD40 subfamily of proteins regulated by salt stress [52], and that salt tolerance of rice seedlings can be enhanced by overexpression of the *OsABT* gene, which encodes a WD40 repeat protein [53].

WD40 protein family is involved in the anthocyanin synthesis pathway. Anthocyanin is favorable for the dietary management of metabolic disorders such as diabetes and hyperlipidemia [54]. And anthocyanin in pigmented rice is beneficial to human health and can also be used as a natural pigment and nutraceutical [55,56]. Additionally, the purple coloration of the non-edible part was used in breeding programs as a morphological marker to identify the varieties and to study the linkage analysis [57]. At present, the anthocyanin synthesis-related WD40 gene in rice has only been cloned and studied by *OsTTG1* [30]. Studies have demonstrated that *OsTTG1* is localized in the nucleus and physically interacts with Kala4, *OsC1*, *OsDFR*, and Rc. There are 59 transcription factors that might be involved in influencing anthocyanin synthesis, and *OsWD40-3* (*LOC_Os01g28680*) might be functionally redundant with *OsTTG1*. In this study, the *WD40* subfamily in the *Oryza* genus was identified and analyzed for its characteristics, sequence structure, linkage, and cis-acting elements. In addition, the expression profiles of *WD40* were analyzed by a combination of RNA-Seq and qRT-PCR to provide further insight into their functions.

Comparative genomics analysis in allied species can greatly improve the understanding of gene evolution. Based on the previous classification of the *WD40* superfamily in japonica, the members of the subfamily in which WD40-TTG1 is located, the *Oryza* genus, were analyzed and validated. A total of 51 members were finally identified in *O. sativa*, *indica*, 48 in *O. nivara*, 48 in *O. rufipogon*, 46 in *O. glaberrima*, 37 in *O. meridionalis*, 48 in *O. barthii*, 46 in *O. glumaepatula*, and 9 in *O. longistaminata*. The 383 members, including *O. sativa* and *japonica*, were divided into four groups based on the topology. The present study enriches the understanding of the *WD40* family in eukaryotes, especially in the *Oryza* genus. Moreover, transposition, segmental duplication, and tandem duplication play an important role in biological evolution [58]. In this study, all gene pairs undergoing duplication events were generated by segmental duplication or WGD duplication, suggesting that segmental duplication or WGD duplication might have a large role in the *Oryza genus*, and they are mainly from subgroups B and C. Therefore, it can be inferred that segmental duplication is an important driver for the expansion of specific subfamilies of *WD40s*. The Ka/Ks ratio can determine whether the genes encoding proteins are subject to selection pressure [59]. The Ka/Ks analysis in this study showed that all duplicated WD40 gene pairs in the AA genome underwent purifying selection, suggesting that the elimination of unfavorable mutations might facilitate adaptation to complex environments in rice.

Genes with similar amino acid sequences will usually have similar functions [60]. In this study, *OsiWD40-15*, a member of the WD40 family in indica, showed the highest similarity to TTG1s of other plants with 61.42%, 99.70%, 58.77%, 61.31%, 61.18%, 61.88%, 61.47%, 59.76% identity to *AtTTG1*, *OsTTG1*, *BvTTG1*, *RsTTG1*, *MdTTG1*, *RrTTG1*, *PyTTG1*, and *PgTTG1*, respectively [61,62,63,64,65]. In addition, *OsiWD40-15* shares four highly conserved WDR motifs with other TTG1s, which might be related to anthocyanin biosynthesis. In most cases, the ternary complex MYB-bHLH-WD40 is an important player in the activation of structural genes for phycocyanin biosynthesis [27]. The protein–protein interactions analysis showed that *OsiWD40-15* might be interacting with MYB and bHLH transcription factors. It can be inferred that *OsiWD40-15* might also be able to participate in the anthocyanin synthesis pathway as a member of the MBW complex. In addition to analyzing the *Oryza* genus, interspecies evolutionary analyses of WD40 could be performed in more gramineous plants. However, some possible protein interactions were not validated in this study. Moreover, WD40 is usually carried out as one of the MBW complex components in the anthocyanin synthesis pathway. The MYB and bHLH transcription factors that might interact with WD40 can be studied in further studies, such as yeast two-hybrid hybridization and subcellular localization. Moreover, relevant candidate genes can be explored for breeding.

## 4. Materials and Methods

### 4.1. Plant Material, Growth, and Stress Conditions

Rice material 9311 (*Indica*, *Oryza sativa*) was cultured in a 26 °C chamber at Wuhan University, Wuhan, China with a light/dark photoperiod of 16/8 h and 60% relative humidity [66]. To study the response of *OsiWD40s* to hormone and abiotic stresses, plants were subjected to different treatments at the three-leaf stage, such as 100 μM Abscisic acid (ABA), 50 μM 3-Indole acetic acid (IAA), 50 μM Gibberellic acid (GA), 100 μM Methyl Jasmonate (MeJA), 120 mM NaCl (salt treatment), 25% polyethylene glycol (PEG 6000, drought treatment), and 5 low temperatures (cold treatment). The reagents used were obtained from Biosharp, Anhui, China, and MACKLIN, Wuhan, China. The solution with the configured concentration was added to the liquid medium respectively, and leaves were collected after 24 h treatments and stored at −80 °C. Three biological replicates were used for each treatment.

### 4.2. Identification and Phylogenetic Analysis of WD40 Genes in Rice

The whole genome datasets of *O. nivara*, *O. rufipogon*, *O. glaberrima*, *O. meridionalis*, *O. barthii*, *O. glumaepatula* and *O. longistaminata* from Ensemble Plant “http://plants.ensembl.org/index.html (accessed on 25 October 2022)”, data of *O. sativa indica*(R498) from MBKbase “https://mbkbase.org/rice (accessed on 25 October 2022)”, and data of *O. sativa japonica* (*Nipponbare*) from Rice Genome Annotation Project “http://rice.uga.edu/ (accessed on 25 October 2022)“. Next, the HMM (Hidden Markov Model) file of WD40 (PF00400) was obtained from the Pfam database “http://pfam-legacy.xfam.org/ (accessed on 25 October 2022)” and used as query to search for eight species using HMMER v3.3.2 [67,68]. Then, 50 *OsWD40s* protein sequences extracted from *O. sativa japonica* were used as query sequences to search the local protein database (E-value of e-5) by BLASTP software (v2.14.1). Then, the results of HMMER and BLASTP methods were intersected [69]. Then, all candidate WD40 protein sequences were tested for the presence of the WD40 structural domain using the hmmscan program of HMMER. All WD40 gene accession names are shown in Appendix A. The identified WD40 protein sequences were subjected to multiple sequence alignment using mafft v7.475 default parameters [70]. Subsequently, phylogenetic trees were constructed using fasttree based on the maximum likelihood method with default parameters [71]. The species tree was obtained from TimeTree “http://www.timetree.org/ (accessed on 29 August 2023)” based on the divergence time [72]. Finally, they were uploaded to iTOL online tool “https://itol.embl.de/ (accessed on 14 June 2023)” for further refinement [73].

### 4.3. Chromosomal Localization, Gene Structure, and Conserved Motifs

To obtain the genomic distribution of the WD40s, chromosomal location and gene structure information of the members were extracted from the GFF3 file and further visualized by TBtools. Then, TBtools was used to provide a comprehensive analysis of the physical and chemical properties of *WD40s*, and amino acid size, isoelectric point, and molecular weight were analyzed with TBtools [74]. In addition, the WD40s protein sequence was uploaded to BUSCA “http://busca.biocomp.unibo.it/ (accessed on 30 August 2023)” for protein subcellular localization prediction [75]. Then, MEME 5.5.0 (parameters: -protein -oc. -nostatus -time 14,400 -mod zoops -nmotifs 10 -minw 6 -maxw 50 -objfun classic -markov_order 0) to identify up to 10 conserved motifs in these proteins [76]. The motif information is shown in Appendix A. Then, motifs were uploaded to Tomtom “https://meme-suite.org/meme/doc/tomtom.html?man_type=web (accessed on 12 October 2023)” and analyzed the sequences in which the motifs might code [77]. Subsequently, a 2000 bp sequence upstream of the WD40 gene was extracted using TBtools, and cis-acting elements were retrieved through the PlantCARE online tool “http://bioinformatics.psb.ugent.be/webtools/plantcare/html/ (accessed on 15 June 2023)”.

### 4.4. Collinearity Analysis of WD40 Genes

Collinearity analysis using the MCScanX toolkit with default parameters to explore *WD40s* duplexing events in AA genome and visualization of collinear gene pairs using TBtools [78]. Ka, Ks, and Ka/Ks ratios of duplicated genes were calculated using TBtools’ simple Ka/Ks calculator. Divergence time (T) was calculated as T = Ks/(2 × 6.5 × 10 ^−9^) × 10 ^−6^ million years ago (Mya) [79].

### 4.5. Expression and Interaction Analysis of OsiWD40 Gene Family

Expression profile data of *OsiWD40s* at different stages in multiple tissues were retrieved from the MBKbase database “https://mbkbase.org/rice (accessed on 21 June 2023)” [80], and pheatmap package of R software (4.1.2) was used to plot heatmap. STRING database “https://cn.string-db.org (accessed on 28 June 2023)” was used to predict the protein-interaction relationship of OsiWD40 [81].

### 4.6. RNA Extraction and qPCR Analysis

Total RNA of the samples was extracted using FastPure Plant Total RNA Isolation Kit. First-strand cDNA was synthesized using UEIris RT mix with DNase (all-in-one), and Hieff UNICON Universal Blue qPCR SYBR Green Master Mix was used for quantitative analysis according to the instructions. Untreated samples were used as controls, and three technical replicates were performed for each gene. qRT-PCR reactions were performed on a CFX96 TouchTM Real-Time PCR Detection System (Bio-Rad, Hercules, CA, USA) with the following thermal cycling conditions: pre-denaturation at 95 °C for 10 min, denaturation at 95 °C for 10 s, annealing at 60 °C for 10 s, and 72 °C extension for 15 s. The reactions were repeated for 40 cycles, and solubility curves were plotted from 65 °C to 95 °C to test the specificity of the primers. The actin gene (UBI) was used as an internal reference control, and the primers used are shown in Appendix A. Relative expression levels were calculated based on three biological replicates using the 2^−∆∆CT^ method [82]. A two-way ANOVA was performed on the data, and the results were visualized in GraphPad Prism9.

## 5. Conclusions

In this study, 383 WD40 subfamily members were identified in the *Oryza* genus and classified into four groups. These first analyzed the evolutionary patterns of the WD40 subfamily in multiple species. They have similar chromosome distribution and physiological characteristics. In addition, segmental duplication was the main expansion pattern during the WD40 gene family evolution, and collinearity was similar in the *Oryza* genus. As an example, the expression analysis in indica showed that WD40 genes might be involved in the growth and development of several tissues and organs, such as roots, stems, leaves, and flowers. Many metabolic elements, hormone response elements, and stress response elements were found in the promoter region. In addition, the expression patterns in indica were analyzed under various phytohormonal and abiotic stresses, and some genes played important roles in the growth and stress tolerance process. These results provide new insights into the functions of the WD40 family in species evolution and rice development and have potential applications in breeding for rice quality improvement.

## Figures and Tables

**Figure 1 ijms-24-15776-f001:**
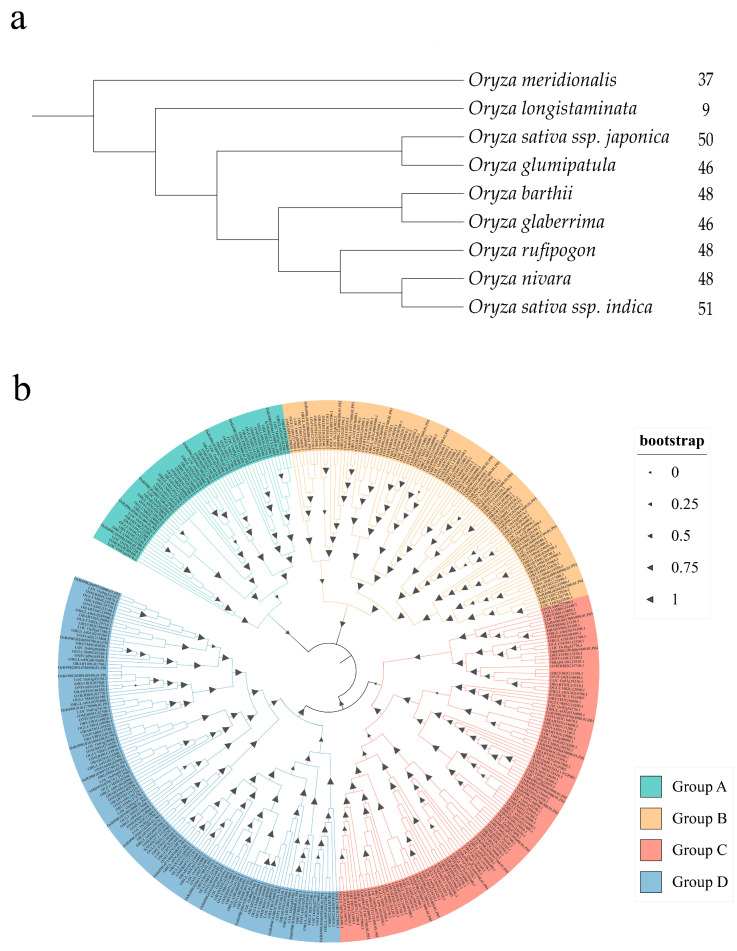
Phylogenetic relationship of WD40 repeat protein family identified in *Oryza* genus. (**a**) Tree based on divergence times and the number of members of the respective family is indicated next to the species name. (**b**) Phylogenetic tree of *Oryza genus* constructed based on the maximum likelihood method, divided into four groups: A, B, C, and D. The triangle size indicates the bootstrap value.

**Figure 2 ijms-24-15776-f002:**
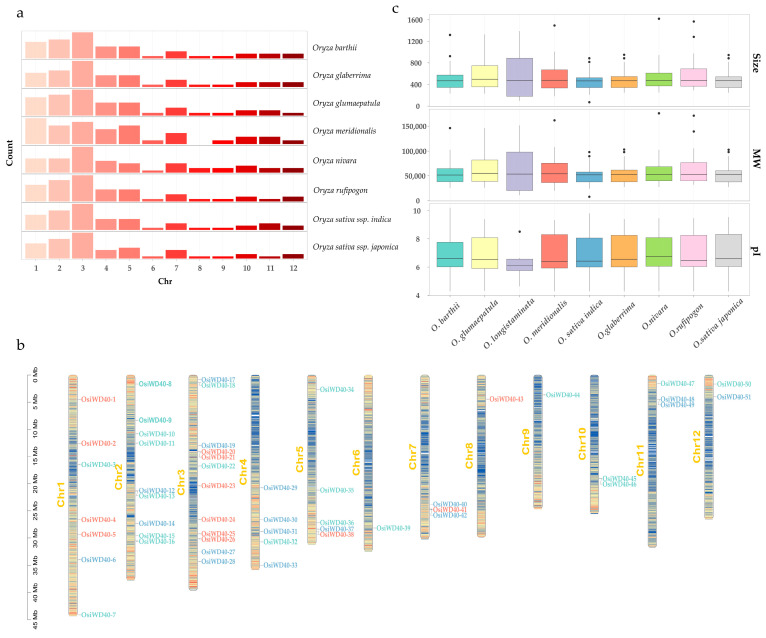
Chromosome distribution of the WD40s in *Oryza* genus. (**a**) The bar chart of the distribution of genes on chromosomes in each species. The *x*-axis is the chromosome number, and the *y*-axis is the gene count. The right side of the bar graph shows the corresponding species name. (**b**) The bars represent chromosomes. Chromosome numbers are shown on the left side of the chromosome. The *OsiWD40* is labeled on the right side of the chromosome. The scale bar on the left side indicates the length of the chromosome. (**c**) The box plots of amino acid size, molecular weight, and isoelectric point in *Oryza* genus. The *x*-axis represents species name, and *y*-axis represents the size of each of the three metrics. The size of box reflects the range of data, the line inside represents the median, and the dots outside the box mean outliers.

**Figure 3 ijms-24-15776-f003:**
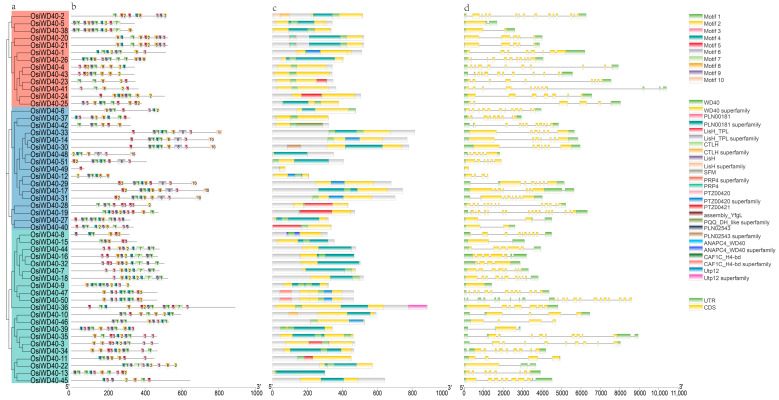
Phylogenetic relationship, conserved motifs, gene structure, and cis-acting factor of *OsiWD40s*. (**a**) An unrooted tree of *OsiWD40s* using fasttree based on protein sequences. The *OsiWD40s* was renamed. (**b**) Conserved motifs of *OsiWD40s* analyzed with MEME. (**c**) Domain prediction using CDD database. (**d**) Introns and exons of the *OsiWD40s*. Note: Different colors indicate different meanings; see the figure legends for details.

**Figure 4 ijms-24-15776-f004:**
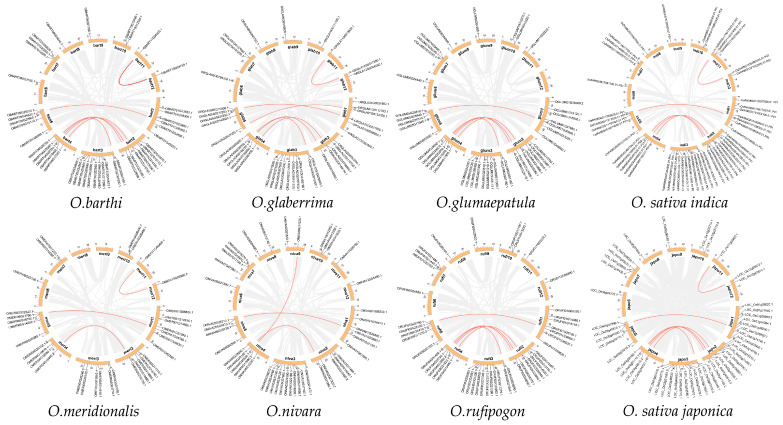
Collinearity analysis of *OsiWD40s* in *Oryza*. Collinear gene pairs are shown for eight species except for *O. longistaminata*. Red lines represent collinear gene pairs, and labels outside the circles are family member numbers for each species.

**Figure 5 ijms-24-15776-f005:**
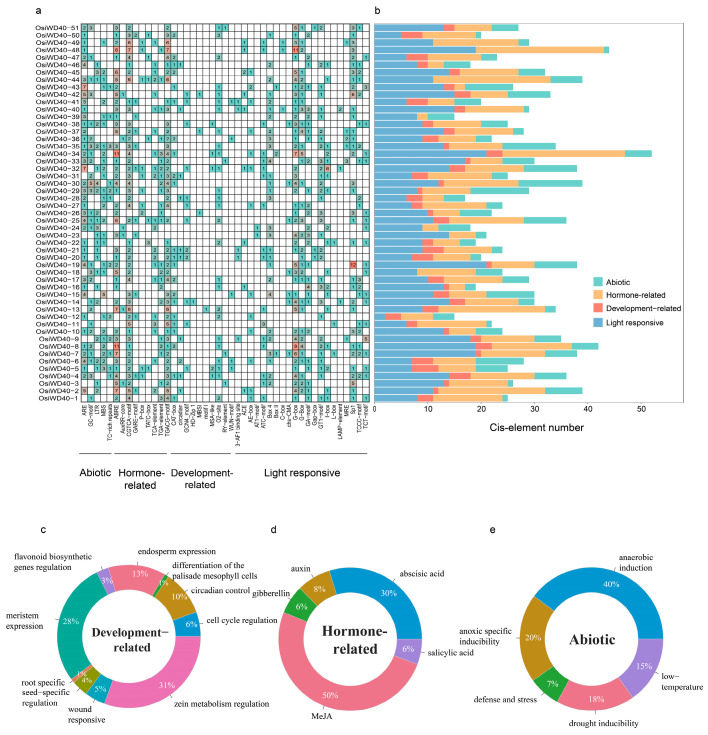
Predicted cis-elements of the *OsiWD40* gene family. (**a**) The number in the box represents the number of corresponding motifs contained in the 2 kb promoter sequence upstream of the gene. Blank indicates that the element is not included, and red and blue indicate the number of elements; the redder the color, the greater the number of elements with a given motif. (**b**) The various motifs are classified into four major categories: abiotic, hormone-related, development-related, and light-responsive, and each colored bar represents the number of element classifications contained in that gene. (**c**–**e**) The proportion of the number of each element in the three major categories of development-related, hormone-related, and abiotic, respectively.

**Figure 6 ijms-24-15776-f006:**
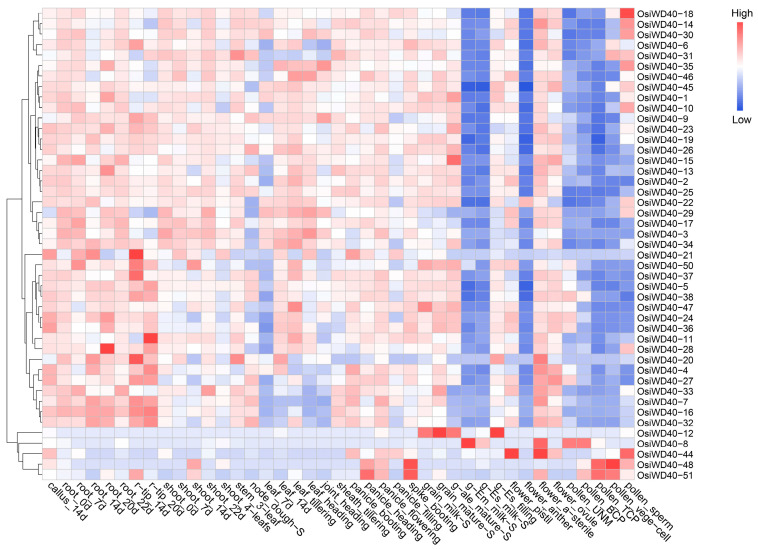
Heat map of the expression of *OsiWD40s*. The redder color of the color block represents higher expression, and the bluer color block represents lower expression.

**Figure 7 ijms-24-15776-f007:**
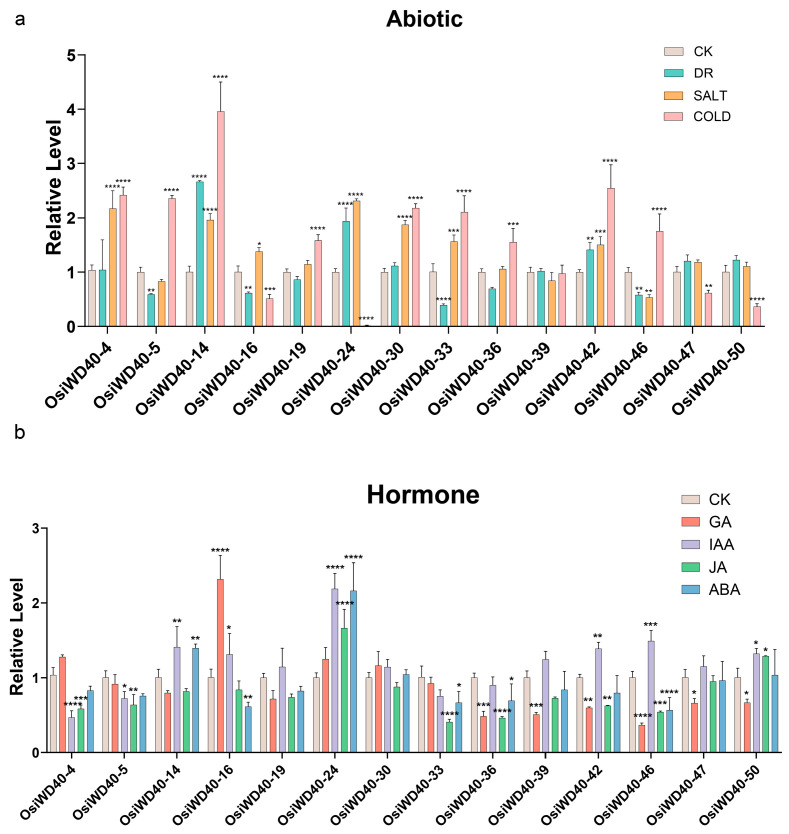
Relative expression level of 14 OsiWD40s in response to abiotic and hormone treatments. Error bars are standard deviations of three biological replicates. Asterisks were used to indicate the significant degree of the expression level compared to control. (* *p* < 0.05, ** *p* < 0.01, *** *p* < 0.001, **** *p* < 0.0001) (**a**) Abiotic treatments. CK: control, DR: drought, SALT: salt, COLD: cold. (**b**) Hormone treatments: CK: control, GA: gibberellic acid, IAA: auxin, JA: methyl Jasmonate, ABA: abscisic acid.

**Table 1 ijms-24-15776-t001:** The Ka/Ks values in duplicated gene pairs in *Oryza* genus.

*Oryza* Species	Gene Pairs	Ka/Ks	Date (Mya)	Type of Selection	Type of Duplication
*O. barthii*	*ObWD40-5/ObWD40-36*	0.24	40.76	Purifying	SD/WGD
	*ObWD40-46* */ObWD40-48*	0.38	45.95	Purifying	SD/WGD
	*ObWD40-13* */ObWD40-28*	0.28	68.50	Purifying	SD/WGD
	*ObWD40-15* */ObWD40-30*	0.09	50.86	Purifying	SD/WGD
	*ObWD40-11* */ObWD40-27*	0.22	133.43	Purifying	SD/WGD
	*ObWD40-16* */ObWD40-29*	0.23	146.98	Purifying	SD/WGD
*O. glaberrima*	*OglaWD40-4* */* *OglaWD40-34*	0.24	34.28	Purifying	SD/WGD
	*OglaWD40-3* */* *OglaWD40-30*	0.17	90.11	Purifying	SD/WGD
	*OglaWD40-43* */* *OglaWD40-45*	0.07	5.86	Purifying	SD/WGD
	*OglaWD40-10* */* *OglaWD40-25*	0.22	129.39	Purifying	SD/WGD
	*OglaWD40-13* */* *OglaWD40-28*	0.09	50.86	Purifying	SD/WGD
	*OglaWD40-14* */* *OglaWD40-27*	0.23	146.07	Purifying	SD/WGD
*O. glumaepatula*	*OgluWD40-5* */* *OgluWD40-35*	0.23	37.97	Purifying	SD/WGD
	*OgluWD40-11* */* *OgluWD40-26*	0.21	131.89	Purifying	SD/WGD
	*OgluWD40-13* */* *OgluWD40-27*	0.23	59.40	Purifying	SD/WGD
	*OgluWD40-15* */* *OgluWD40-29*	0.14	56.44	Purifying	SD/WGD
	*OgluWD40-16* */* *OgluWD40-28*	0.23	145.73	Purifying	SD/WGD
*O. indica*	*OsiWD40-5* */* *OsiWD40-38*	0.27	33.55	Purifying	SD/WGD
	*OsiWD40-12* */* *OsiWD40-29*	0.12	125.12	Purifying	SD/WGD
	*OsiWD40-14* */* *OsiWD40-30*	0.23	60.94	Purifying	SD/WGD
	*OsiWD40-16* */* *OsiWD40-32*	0.09	49.85	Purifying	SD/WGD
	*OsiWD40-47* */* *OsiWD40-50*	0.08	5.03	Purifying	SD/WGD
*O. meridionalis*	*OmWD40-5* */* *OmWD40-27*	0.23	58.16	Purifying	SD/WGD
	*OmWD40-36* */* *OmWD40-37*	0.53	56.54	Purifying	SD/WGD
	*OmWD40-11* */* *OmWD40-19*	0.25	126.22	Purifying	SD/WGD
*O. nivara*	*OnWD40-4* */* *OnWD40-32*	0.22	144.04	Purifying	SD/WGD
	*OnWD40-12* */* *OnWD40-27*	0.10	34.58	Purifying	SD/WGD
	*OnWD40-26* */* *OnWD40-41*	0.25	41.55	Purifying	SD/WGD
*O. rufipogon*	*OrWD40-5* */* *OrWD40-37*	0.23	52.06	Purifying	SD/WGD
	*OrWD40-14* */* *OrWD40-29*	0.23	43.46	Purifying	SD/WGD
	*OrWD40-16* */* *OrWD40-31*	0.14	129.31	Purifying	SD/WGD
	*OrWD40-12* */* *OrWD40-28*	0.23	115.74	Purifying	SD/WGD
	*OrWD40-17* */* *OrWD40-30*	0.23	123.19	Purifying	SD/WGD
*O. japonica*	*OsWD40-5* */* *OsWD40-37*	0.23	34.98	Purifying	SD/WGD
	*OsWD40-47/OsWD40-48*	0.08	4.48	Purifying	SD/WGD
	*OsWD40-14* */* *OsWD40-30*	0.26	64.56	Purifying	SD/WGD
	*OsWD40-16* */* *OsWD40-31*	0.10	49.85	Purifying	SD/WGD
	*OsWD40-12* */* *OsWD40-29*	0.22	129.57	Purifying	SD/WGD
	*OsWD40-17* */* *OsWD40-30*	0.21	155.10	Purifying	SD/WGD

## Data Availability

Data are contained within the article or Appendix A.

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
