# Peer review of "Genome-Wide Identification, Evolution, and Expression Analysis of the WD40 Subfamily in Oryza Genus"

_ijms, 2023, doi:10.3390/ijms242115776_

Round 1

Reviewer 1 Report (New Reviewer)

Comments and Suggestions for Authors

Dear Authors,

Ke et al. in their study, they discuss the role and significance of the WD40 protein family, especially among rice and closely related plants. The study identifies and analyzes the presence and evolution of WD40 genes and presents their expression patterns under different conditions. Overall, the manuscript has serious shortcomings and needs improvement in several areas to be scientifically acceptable. The text does not begin to clearly present the background of the research or the main objectives. A brief introduction or background information could help the reader better understand the need for the research. Connections: The text does not present the background and importance of the research, how the results are related to the wider research areas or to the agronomic aspects of the rice plant. Lack of purpose and context: The abstract and Introduction do not clearly outline the purpose or context of the research. It would be important for readers to know why the WD40 protein family is being investigated and what prior knowledge the research is based on. Unstructured content: The manuscript is unstructured and difficult to follow. There is a lack of consistent addressing and logical progression of the text. Lack of critical analysis: The text does not contain a critical analysis or evaluation of the limitations of the research or the possibilities of further research arising from the results. However, these shortcomings and errors can be corrected! Overall, the text could function as a much more advanced scientific paper with the above corrections and additions. There are several grammatical and spelling errors in the text that also need to be corrected.

Comments on the Quality of English Language

There are several grammatical and spelling errors in the text that also need to be corrected.

Author Response

Dear editors,

We really appreciate that you give us many useful advices. And we have made corresponding revision point by point. The supplement word file is the manuscript we had revised.

Point 1:  The text does not begin to clearly present the background of the research or the main objectives. A brief introduction or background information could help the reader better understand the need for the research.

Response 1: Based on the journal template, we don't know if we can add the backgrougnd section. So we figured out how to make line8-12 a short background in Abstract section, which we modified as you suggested. And in Introduction section, we adjusted the logical order. In this section, the structure of WD40 protein, the number of members of the WD40 protein in multiple species that have been identified to date, and the various important functions that have been discovered are presented to demonstrate the importance of WD40 protein family. It also leads to the fact that the Oryza genus is an ideal research system in graminaceous plants and that WD40 protein family and its evolutionary relationships have not been studied in the Oryza genus, i.e., it clarifies the reasons why we should study the WD40 protein family in the Oryza genus.

Point 2:  Connections: The text does not present the background and importance of the research, how the results are related to the wider research areas or to the agronomic aspects of the rice plant. 

Response 2: We have added the statements in Abstract(line 30-33), Introduction(line 123-126), and Discussion(line 340-345) sections that discovered candidate genes can be used for subsequent breeding and morphological marker development.

Point 3:  Lack of purpose and context: The abstract and Introduction do not clearly outline the purpose or context of the research. It would be important for readers to know why the WD40 protein family is being investigated and what prior knowledge the research is based on. Unstructured content: The manuscript is unstructured and difficult to follow.  There is a lack of consistent addressing and logical progression of the text.

Response 3: We have organized the Introduction section to provide specific background by WD40 structure - WD40 family members – WD40 functions. Then, we introduced the Oryza genus, which is the subject of our study in this manuscript, and suggested that there has been no study of the WD40 protein family in the Oryza genus. In the context of the above, we hope we have elucidated why the WD40 protein family should be studied in the Oryza genus.

Point 4:  Lack of critical analysis: The text does not contain a critical analysis or evaluation of the limitations of the research or the possibilities of further research arising from the results.

Response 4: In discussion section, we analyzed and explored some experimental validation of possible protein interactions that could be done in future study. It is pointed out that relevant molecular experiments can be carried out for the candidate genes in the future.

Point 5:  There are several grammatical and spelling errors in the text that also need to be corrected.

Response 5: We have checked our manuscript carefully, and corrected several grammatical and spelling errors.

Thank you very much for your kindly advice. 

Best regards,

Simin Ke

Reviewer 2 Report (New Reviewer)

Comments and Suggestions for Authors

In this manuscript, Ke and collaborators performed a genome-wide identification and analysis of WD40 subfamily comprising OsTTG1 in rice (O. sativa) and identify putative orthologs in other 7 related species, even if you concentrated your analyses only one subspecies, indica. Given this, the title does not seem to match fully what has been done in the paper, since no mention was given about the subfamily or the other-than-O.sativa species analysed. In this sense I would modify it to be more adherent to what you presented in your manuscript.

As in similar papers, the manuscript provides a broad and raw analysis of the whole family and hence a starting step for more in-depth analyses of members of WD40 proteins. Besides providing in silico analyses, the authors perform qPCR in order to analyse the expression levels of candidate WD40 subfamily OsTTG1 and their response to different stimuli, and this is appreciable since the work does not only rely on in silico analyses.

I have some questions that I ask you to consider, here below

The paper however needs some check as in some parts it is not clear what the authors want to indicate (as for instance in lines 144-149). Moreover, some panels in Figures are not described or mentioned in the text (eg Fig 3a, 3b)

I'd request some additional info about the genome of O. longistamata you used for the analyses: if it has not been annotated yet (lines 139-140), how have you got your WD40 proteins? If you annotated yourselves or used other annotations, you have to describe it in the appropriate Material & Methods section. If another approach was used, could this have an impact to downstream analyses?

In the manuscript you analysed both indica and japonica subspecies: have you found any difference regarding structure, promoter region, etc?

You used PlantCARE to analyze WD40 subfamily promoter region for cis-acing elements and performed qPCR using leaf under hormone and abiotic treatments: could you see a relationship with expression level and CREs?

section 2.3: did you check whether the conserved motifs identified by MEME actually do code for something?

section 2.7: here you describe that genes were significantly up- or down-regulated, could you provide a description or statistics supporting this? 

section 4.1: please add some detail about the treatments (such as their duration).

other than that, some smaller things

line 119: I guess you mean the results of ref 39 but it is not very clear, please rephrase

lines 144-145 and 147-148, please rephrase as it is not clear what you did

line 214: it is not positive but purifying selection

In order to normalize qPCRs, you used only one actin gene; how did you check that the primers were universal, that is, could amplify also in other species?

panel c of Figure 2 is missing in the caption

Comments on the Quality of English Language

Please revise the manuscript as some phrases seem to be either incomplete or with unnecessary stops (eg.: lines 101, 107, ...), repetitions (eg.: lines 391-393,...) 

Author Response

Dear editor,

We feel great thanks for your professional review work on our article. As you are concerned, there are several problems that need to be addressed. According to your nice suggestions, we have made extensive corrections to our previous draft, the detailed corrections are listed below.

Point 1: In this manuscript, Ke and collaborators performed a genome-wide identification and analysis of WD40 subfamily comprising OsTTG1 in rice (O. sativa) and identify putative orthologs in other 7 related species, even if you concentrated your analyses only one subspecies, indica. Given this, the title does not seem to match fully what has been done in the paper, since no mention was given about the subfamily or the other-than-O.sativa species analysed. In this sense I would modify it to be more adherent to what you presented in your manuscript.

Response 1: We have changed the title from “Oryza sativa” to “Oryza genus” as an indication that the species studied in this manuscript.

Point 2: The paper however needs some check as in some parts it is not clear what the authors want to indicate (as for instance in lines 144-149). Moreover, some panels in Figures are not described or mentioned in the text

Response 2: We wanted to show the relationship between chromosome distribution and chromosome length. And we have reorganized the statements of line144 and added the descriptions of Fig3a, 3b to the text.

Point 3: I'd request some additional info about the genome of O. longistamata you used for the analyses: if it has not been annotated yet (lines 139-140), how have you got your WD40 proteins? If you annotated yourselves or used other annotations, you have to describe it in the appropriate Material & Methods section. If another approach was used, could this have an impact to downstream analyses?

Response 3: Its gene and protein sequences are accessible now, but its genomic information has not yet been assembled and mounted on the corresponding chromosome. Only scaffold information can be obtained now. So now it's possible to analyze their genome or protein sequences, but there's no way to map them to a specific chromosome.

Point 4: In the manuscript you analysed both indica and japonica subspecies: have you found any difference regarding structure, promoter region, etc?

Response 4: In Figure S1, we analyzed the conserved motifs and structural domains of all family members in Oryza genus, and found that WD40 subfamily proteins in Oryza genus are always interspersed in the phylogenetic tree. This was mentioned at the beginning of 2.3(line 186-191).

Point 5: You used PlantCARE to analyze WD40 subfamily promoter region for cis-acing elements and performed qPCR using leaf under hormone and abiotic treatments: could you see a relationship with expression level and CREs?

Response 5: We have added part of analysis that combines the expression level and CREs. And there are also many factors that affect the level of gene expression, and we have added a little correlation to the analysis as appropriate.

Point 6: section 2.3: did you check whether the conserved motifs identified by MEME actually do code for something?

Response 6: After referring to "Quantifying similarity between motifs" (doi: 10.1186/gb-2007-8-2-r24), we further analyzed the motifs and found that motifs 1-4,9 have WD40 repeat signature. We have added this part into section 2.3 and added the reference method in section 4.3.

Point 7: section 2.7: here you describe that genes were significantly up- or down-regulated, could you provide a description or statistics supporting this? 

Response 7: We performed a two-way ANOVA on the qPCR result data and labeled the significance results in the figure and legend.

Point 8: section 4.1: please add some detail about the treatments (such as their duration).other than that, some smaller things

Response 8: We added specific processing details as well as duration in 4.1.

Point 9: line 119: I guess you mean the results of ref 39 but it is not very clear, please rephrase

Response 9: We added a reference at that location to indicate that the result from ref 39. However, we have some additional changes in Introduction section, so the ref number here has changed to 18.

Point 10: lines 144-145 and 147-148, please rephrase as it is not clear what you did

Response 10: We reorganized the sentences here to illustrate the relationship between chromosome distribution of family genes and chromosome length.

Point 11: line 214: it is not positive but purifying selection

Response 11: We have corrected the error here.

Point 12: In order to normalize qPCRs, you used only one actin gene; how did you check that the primers were universal, that is, could amplify also in other species?

Response 12: Actin gene is a commonly used reference gene in rice, and its expression is relatively stable across tissues in periods. We designed qPCR primer sequences for it and analyzed the solubility curves for all the primer sequences designed for the gene, which resulted in a single peak proving to be a specific amplification product. And the Ct value of the internal reference gene was stable around 22-23 in different samples.

Point 13: panel c of Figure 2 is missing in the caption

Response 13: We have added the caption of Fig2c.

Point14:Please revise the manuscript as some phrases seem to be either incomplete or with unnecessary stops (eg.: lines 101, 107, ...), repetitions (eg.: lines 391-393,...) 

Response14:We checked for errors in grammar or usage that appeared in the manuscript.

We would like also to thank you for allowing us to resubmit a revised manuscript.

Sincerely,

Simin Ke

Round 2

Reviewer 1 Report (New Reviewer)

Comments and Suggestions for Authors

Dear Authors,

Thank You for your cooperation. And thank You for your detailed answer, I consider your manuscript accepted in present form.

Best Regards,

Reviewer

.

Author Response

Dear reviewer,

We really appreciate your valuable time and patience.

Sincerely,

Simin Ke

Reviewer 2 Report (New Reviewer)

Comments and Suggestions for Authors

The authors provided a quite improved manuscript in its 2nd version; save some revision concerning English and some minor spelling and formating issue that will be fixed at later stages, I feel the manuscript can be accepted for publication.

Comments on the Quality of English Language

As in the last version, please check the verb tense (eg.: lines 187 and 191). I would also rephrase lines 191-194 as, if I unerstood correctly, you extended to the Oryza genus what has been done in the sativa sub japonica.

Please check carefully also the modifications you included in this v2 version.

Author Response

Dear editor,

We apologize for the omission of verb tense check in revision process.

We have checked the entire manuscript again, and rephrased line 191-194.

All corrections including wrong tense in manuscript were highlighted in yellow.

Thanks again for your patience!

Sincerely,

Simin Ke

This manuscript is a resubmission of an earlier submission. The following is a list of the peer review reports and author responses from that submission.

Round 1

Reviewer 1 Report

Comments and Suggestions for Authors

This manuscript describes the evolution of the WD40 family in the rice genus, including the comprehensive identification of TTG1 homolog WD40 in nine species and their phylogenetic and evolutionary analysis. The data provide a basis for elucidating the function of TTG1 homologs in the Oryza genus. However, the manuscript has several issues, as listed below. 

1) The specific physiological function of the TTG1 homolog has not been clarified, and localization and expression analysis of the cis-element has only been performed in indica rice.

2) It is unclear what the most important new finding is. In the "Introduction," please clearly state the purpose of this paper. Also, in the "Conclusion," please clearly state the novel findings.

For example, lines 25-26 describe, "These results provide a rich perspective for exploring the evolution of the WD40 subfamily and its practical application in rice." However, the results do not provide a perspective for practical application because this manuscript does not show the biological and physiological function of the WD40 subfamily. Lines 386-389 describe that "our results provide valuable insights into the functional diversity", although this manuscript does not show the biological and physiological function of the WD40 subfamily. 

3) In the phylogenetic tree in Figure 1A, O. nivara and O. rufipogon are clades separated from O. sativa. It is the reviewer's understanding that these species are closely related (Zhu et al., 2014; https://doi.org/10.1016/j.ympev.2013.10.008). For the benefit of readers unfamiliar with the evolution of the Oryza genus, please briefly describe the species used in the manuscript in the “Introduction.” In particular, it is crucial whether it is closely related to O. sativa, and whether it is wild or cultivated rice.

Author Response

Dear sir,

We feel great thanks for your professional review work on our article. As you concerned, there are several problems that need to be addressed.according to you suggestions, we have made extensive corrections to our previous draft, the detailed corrections are listed below.

Point 1: The specific physiological function of the TTG1 homolog has not been clarified, and localization and expression analysis of the cis-element has only been performed in indica rice.

Response 1:We have added a section on TTG1 functionality to the Introduction section, which can be found in lines 74-82 of the Annex. And due to the large number of members in this subfamily and the fact that all members of WD40 in japonica rice have been identified in the literature(Ouyang et al. BMC Genomics 2012, http://www.biomedcentral.com/1471-2164/13/100), we chose indica rice as the model plant for cis-acting element analysis, which is more convenient for the presentation of the results and is also representative.

Point 2:It is unclear what the most important new finding is. In the "Introduction," please clearly state the purpose of this paper. Also, in the "Conclusion," please clearly state the novel findings.

Response 2:We have not been precise enough in our presentation, so we have revised the Abstract(lines 8-11), Introduction(lines 90-100) and Conclusion(lines 393-404) sections of the article to clarify the new findings and the purpose of this paper.

Point 3:In the phylogenetic tree in Figure 1A, O. nivara and O. rufipogon are clades separated from O. sativa. It is the reviewer's understanding that these species are closely related (Zhu et al., 2014; https://doi.org/10.1016/j.ympev.2013.10.008). For the benefit of readers unfamiliar with the evolution of the Oryza genus, please briefly describe the species used in the manuscript in the “Introduction.” In particular, it is crucial whether it is closely related to O. sativa, and whether it is wild or cultivated rice.

Response 3:We checked the data and replotted the Figure1a. And we have added some introduction about the AA genome Oryza species, see lines 84-88.

Thank you very much for your attention and time.

Reviewer 2 Report

Comments and Suggestions for Authors

Dear authors! Thank you for submitting the manuscript and the work done. The article is devoted to a fashionable topic, built on the use of modern research methods. The article has some shortcomings: 1. There are outdated references in the bibliography. So out of 51 sources of literature, only 19 references are for 2020-2022. More fresh references for the last 3 years should be added. 2. The article would benefit qualitatively if it also provided information on changes in the level of WD40 expression under biotic stress. In the absence of experimental data on this issue, information from the literature can be added to the Discussion. 3. In Figure 7, there is not enough data on statistical processing of primary data. What is presented in the form of diagrams, data of a typical experiment or average values? Clarify please. 4. In the Materials and Methods section, you should include information about the manufacturers of the reagents you used, such as Abscisic acid, Indole acetic acid, Gibberellic acid, Methyl Jasmonatez, and other reagents. 5. As a whole, the text needs to be proofread, because missing spaces occur (for example, line 328 "v3.3.2[38,39].").

I believe that the article can be successfully published after the comments are removed. Respectfully Yours, reviewer. July 29, 2023

Author Response

Dear sir,

We sincerely thank you for your valuable feedback that we have uesed to improve the quality of our manuscript. And the detailed corrections are listed below.

Point 1: There are outdated references in the bibliography. So out of 51 sources of literature, only 19 references are for 2020-2022. More fresh references for the last 3 years should be added.

Response 1:We have newly cited literature from the last three years at appropriate places, which has now grown to nearly 30 articles.

Point 2: The article would benefit qualitatively if it also provided information on changes in the level of WD40 expression under biotic stress. In the absence of experimental data on this issue, information from the literature can be added to the Discussion. 

Response 2:The time for additional biotic stress experiments may be a bit tight at the moment, but we have added other functionally relevant studies of the WD40 gene by other researchers in the Discussion section(lines 292-297).

Point 3: In Figure 7, there is not enough data on statistical processing of primary data. What is presented in the form of diagrams, data of a typical experiment or average values? Clarify please.

Response 3:After we got the qRT-PCR experimental data with the Bio-rad instrument companion software, we processed the data according to the 2-∆∆CT method and plotted the data statistically with Graph Pad Prism 9.(see 4.6, lines 391-398)

Point 4: 4. In the Materials and Methods section, you should include information about the manufacturers of the reagents you used, such as Abscisic acid, Indole acetic acid, Gibberellic acid, Methyl Jasmonatez, and other reagents.

Response 4:The hormones and other reagents we used were mainly purchased from Biosharp and MACKLIN and have been added to the material(lines 338-341).

Sincerely.

Reviewer 3 Report

Comments and Suggestions for Authors

Dear sir,

the paper 'Genome-Wide Identification, Evolution, and Expression Analysis of the WD40 Subfamily in Rice (Oryza sativa)' deals with the analysis of a protein family WD40. It is a preliminary study to this protein family and their response to phytohormones and abiotic stresses. I don´t see clear the results obtained by the authors, and I have to focus to understand to where the study leads.

Several comments are in the annotated uploaded file. Italics naming are the main mistakes.

Best regards

Comments on the Quality of English Language

Dear sir,

the paper 'Genome-Wide Identification, Evolution, and Expression Analysis of the WD40 Subfamily in Rice (Oryza sativa)' deals with the analysis of a protein family WD40. It is a preliminary study to this protein family and their response to phytohormones and abiotic stresses. I don´t see clear the results obtained by the authors, and I have to focus to understand to where the study leads.

Several comments are in the annotated uploaded file. Italics naming are the main mistakes.

Best regards

Author Response

Dear sir,

Thank you for your constructive comments on this article in your busy schedule. We have carefully read the comments that you have given. The following is my revisions. In addition, we have resubmitted a new manuscript in the reviesed state, with the revisions highlighted in yellow.

Point 1: Several comments are in the annotated uploaded file. Italics naming are the main mistakes.

Response 1:We have corrected the marked areas one by one, please see the areas marked in yellow in the Annex for details.

Thanks for your careful checks.

Best wishes!

Round 2

Reviewer 1 Report

Comments and Suggestions for Authors

The manuscript has not addressed the issues raised in the previous review. Therefore, the reviewer cannot recommend the acceptance of the manuscript.

1) The reviewer pointed out that the manuscript lacks to reveal the physiological function. The revised manuscript still has the same issue. The revised abstract contains too speculative suggestions, such as lines 23-24, 28-30, and 30-32.

(Lines 23-24) The existence of the cis-acting elements related to flavonoid biosynthesis or MYB binding sites does not directly show the involvement of anthocyanin biosynthesis. The functional analysis of each WD40 to show the relation to anthocyanin biosynthesis is needed.

(Lines 28-30) The responsive expression to phytohormones and stress treatment does not directly show the involvement in stress resistance. The functional analysis of each WD40 to show the relation to stress resistance is needed, similar to the above.

(Lines 30-32) The “direction for further exploration of practical applications” is unclear. The manuscript should clearly state the concrete way for the application provided by the results of the manuscript.

2) The revised manuscript is still unclear on purpose and why the comparative genomics using AA genome Oryza species. The revised manuscript describes that “functional diversity were inferred (line 112) “. However, the functional diversity is not shown in the manuscript because several analyses are performed only in indica rice. The expression analysis or functional analysis using other AA genome Oryza species should be performed to show the functional diversity of TTG1 homologs in AA genome species.

3) Why are the Oryza sativa Japonica and Indica distant in the revised Figure 1a? How is the tree in Figure 1a constructed? Please state in detail in the “Materials and Methods”.

Why is the tree different from the relationship in the AA genome Oryza species? The reason should be discussed in the manuscript.

Author Response

Dear sir,

We apologize for not addressing the issues from our last review and appreciate your further corrections to our manuscript. We hope that we can get another chance with this amendment.

Point 1: The reviewer pointed out that the manuscript lacks to reveal the physiological function. The revised manuscript still has the same issue. The revised abstract contains too speculative suggestions, such as lines 23-24, 28-30, and 30-32.

Response 1:

The analysis of this subfamily in the article is partly based on predictions, and we have done qRT-PCR analysis of the genes on rice leaves at the three-leaf stage to verify their expression, which can be used as a reference for the degree of response of these genes to the stresses. It will take a long time to plant the rice material and wait for it to grow up, and it may be a bit tight to meet the deadline as of now. This manuscript is part of my project, mainly to study the evolutionary relationship of the WD40 subfamily, while we will follow up with more experiments such as the verification of anthocyanin-related functions and put it in the future manuscripts to reflect.

(line23-24) The abstract section should reflect the exact results of our experiments, and we would like to use some colored rice varieties with anthocyanin accumulation for further study in the future, so we will put the genes that may be involved in the anthocyanin synthesis pathway in the future manuscript for further functional validation to clarify the conclusions.

(line28-30) Phytohormones play important roles in plant development. In order to understand the possible response of WD40 subfamily to different hormones and abiotic stress treatments, we validated its qRT-PCR for various types of hormones and response to drought and salt stresses, respectively, and found that different genes responded to various types of hormones, drought stresses, and salt stresses to different extents. We will also add more relevant functional analyses in future experiments.

(line30-32) Added practical research directions (line26-28).

These findings contribute to a better understanding of the evolution of the WD40 subfamily, and the analyzed candidate genes can be used for exploring practical applications in rice such as cultivation of colored rice, salt- and drought-tolerant varieties, and development of morphological markers.

Point 2:The revised manuscript is still unclear on purpose and why the comparative genomics using AA genome Oryza species. The revised manuscript describes that “functional diversity were inferred (line 112-) “. However, the functional diversity is not shown in the manuscript because several analyses are performed only in indica rice. The expression analysis or functional analysis using other AA genome Oryza species should be performed to show the functional diversity of TTG1 homologs in AA genome species.

Response 2:Due to the completion of the wild rice genome mapping, the Oryza genus is increasingly becoming an ideal system for evolutionary and functional genomics studies in gramineous plants. The AA genome, however, is the genome type of the rice genus that contains the largest number of diploid species, has the widest distribution, has the most recent origin, is distributed in both natural and wild environments, and contains many genes that can improve rice yield and stress tolerance. Therefore, we chose the AA genome Oryza species for our study to analyze the evolutionary relationships of the WD40 subfamily.

Whereas the WD40 protein family has been identified in Japonica rice in previous studies, andother articles on similar protein family studies have also been referenced (Yuan et al. Genes 2021, https://doi.org/10.3390/genes12050634). Furthermore, Oryza sativa ssp. Indica and Oryza sativa ssp. japonica  are also the most widely studied rice varieties. So indica rice was chosen as the model plant to be analyzed in the subsequent in-depth analysis.

We have added it to the Introduction section(line 82-89,line 93-98) of the article and have yellow labeled it.

Point 3:Why are the Oryza sativa Japonica and Indica distant in the revised Figure 1a? How is the tree in Figure 1a constructed? Please state in detail in the “Materials and Methods”.

Response 3:Figure1a is a tree constructed from the TimeTree database (http://www.timetree.org/) based on divergence time, which has been added to "Materials and Methods"(line360-361). I exported the newick file from the database, refined it on iTOL, and did an "Ignore Branch length" operation on the tree with iTOL to get Figure 1a in the manuscript. The database estimates the most recent common ancestor of different species by aggregating a large amount of primary literature, including extensive manual verification, and constructs phylogenetic tree for the entire population of all accessible species.

Besides, the domestication history of rice has been controversial, and according to Cheng et al. Scientific Reports 2019, https://doi.org/10.1038/s41598-019-47318-x, indica and japonica rice have higher Fst indexes than indica and wild rice, and indica and japonica rice have different selection loci, which suggesting that they experienced different domestication processes and that japonica rice may have experienced a bottleneck event. And it can also be found in Yuan et al. BMC Genomics 2017, https://doi.org/10.1186/s12864-017-3702-x that indica and japonica rice are at a certain distance from each other.

We would really appreciate the opportunity to publish in your journal. We have also revised the manuscript to the best of our ability and will pay more attention to the writing of the manuscript and the logic of the experimental design in the future.

Best regards.

Reviewer 3 Report

Comments and Suggestions for Authors

Dear sir,

It seems that the corrections applied by the authors are ok. To me, it may proceed with publication.

Best regards

Comments on the Quality of English Language

Dear sir,

It seems that the corrections applied by the authors are ok. To me, it may proceed with publication.

Best regards

Author Response

Dear editor,

We appreciate your review, and we'll be double-checking for minor formatting errors in the future!

Best regards.

Round 3

Reviewer 1 Report

Comments and Suggestions for Authors

As pointed out in previous reviews, although the manuscript provides the basis for elucidating the function of TTG1 homologs in the Oryza genus, the physiological function of WD40 is not demonstrated, and this manuscript's novel and important findings are unclear. Therefore, in previous reviews, the reviewer has made the recommendation to reject this manuscript two times. Since this situation has not changed in this revision, the reviewer cannot endorse the acceptance of the manuscript unless the specific physiological function of the TTG1 homolog has been shown. However, the reviewer understands that the final decision is left to the editor. The reviewer will not re-evaluate the manuscript further, leaving the final decision to the editor.